# Preclinical and Clinical Trials of New Treatment Strategies Targeting Cancer Stem Cells in Subtypes of Breast Cancer

**DOI:** 10.3390/cells12050720

**Published:** 2023-02-24

**Authors:** Natalia Landeros, Iván Castillo, Ramón Pérez-Castro

**Affiliations:** 1Unidad de Innovación en Prevención y Oncología de Precisión, Centro Oncológico, Facultad de Medicina, Universidad Católica del Maule, Talca 3480094, Chile; 2In Vivo Tumor Biology Research Facility, Centro Oncológico, Facultad de Medicina, Universidad Católica del Maule, Talca 3480094, Chile; 3Biomedical Research Labs, Facultad de Medicina, Universidad Católica del Maule, Talca 3480094, Chile; 4Hospital Regional de Talca (HRT), Talca 3480094, Chile

**Keywords:** breast cancer, biomarker, metastasis, breast cancer stem cells, stemness

## Abstract

Breast cancer (BC) can be classified into various histological subtypes, each associated with different prognoses and treatment options, including surgery, radiation, chemotherapy, and endocrine therapy. Despite advances in this area, many patients still face treatment failure, the risk of metastasis, and disease recurrence, which can ultimately lead to death. Mammary tumors, like other solid tumors, contain a population of small cells known as cancer stem-like cells (CSCs) that have high tumorigenic potential and are involved in cancer initiation, progression, metastasis, tumor recurrence, and resistance to therapy. Therefore, designing therapies specifically targeting at CSCs could help to control the growth of this cell population, leading to increased survival rates for BC patients. In this review, we discuss the characteristics of CSCs, their surface biomarkers, and the active signaling pathways associated with the acquisition of stemness in BC. We also cover preclinical and clinical studies that focus on evaluating new therapy systems targeted at CSCs in BC through various combinations of treatments, targeted delivery systems, and potential new drugs that inhibit the properties that allow these cells to survive and proliferate.

## 1. Introduction

### 1.1. Breast Cancer: General Aspects

Breast cancer (BC) is the most diagnosed cancer and the leading cause of cancer death in women worldwide [1]. BC has a worldwide incidence of 2,261,419 new cases per year (11.7% of all cancers) and 684,996 deaths (6.9% of all cancer-related deaths) in 2020 [2]. Rates of BC incidence and mortality rates in women are primarily influenced by geographic location and socioeconomic status. High-income North America, Oceania, and Western Europe countries have higher BC incidence rates, while lower-middle-income countries such as South America, East Africa, and Central Asia report fewer women diagnosed with BC but higher mortality, mainly due to late diagnosis and lack of health care resources [3].

BC is considered a complex and multifactorial disease, that arises from the accumulation of multiple genetic alterations and environmental factors. Six of the most relevant risk factors that can increase the possibility of developing BC are: (i) gender, most BC occur in women, only 1% of all BC cases occur in men [4]; (ii) age, the incidence rises sharply as people aged; (iii) reproductive factors such as early menarche, late menopause, late age at first pregnancy and low parity [5,6]; (iv) estrogen, both endogenous and exogenous are associated with the risk of this cancer; (v) modern lifestyles including excessive alcohol consumption and excessive fat intake in the diet, can increase the risk of breast cancer [7], and (vi) family history, almost a quarter of all cases of BC are related to family history, where women, whose mother or sister have BC, are prone to this disease [7,8]. 

The mammary gland is a highly dynamic organ that undergoes multiple phases of remodeling [3]. It is composed of an epithelium containing luminal and basal epithelial cells. The luminal cells form the ducts of the mammary gland, while the basal cells surround the luminal cells, and are in contact with the basal membrane [9]. This apparently hierarchical organization depends on a variety of stem and progenitor cells that populate the mammary gland [3]. The World Health Organization has defined 21 histological types of BC [10]. Breast adenocarcinomas more common are of two types: one that begins in the cells of the ducts (ductal adenocarcinoma) or in the lobules (lobular adenocarcinoma) (Figure 1), which can be in situ, if it has not spread, or invasive (infiltrating) if it has invaded other surrounding breast tissue. In addition, invasive carcinomas are designated by their architecture, secretion (mucinous/colloid), or structural form (medullary, tubular, papillary). Infiltrating ductal carcinoma is the most frequent, representing between 70 to 80% of invasive breast tumors [11]. These subtypes differ in their rate of relapse, metastasis, response to therapies, and composition of cancer cells.

BC is a heterogeneous disease that presents considerable diversity at the molecular, histological, and clinical levels. The first immunohistochemical biomarkers which have laid the foundation for the classification of the BC subtypes mainly include the expression of the estrogen receptor (ER), the progesterone receptor (PR), and human epidermal growth factor receptor 2 (HER2). The evaluation of the expression of ER, PR, and HER2 is a routine procedure in clinical practice, and it is essential for the determination of these subtypes, and so to guide hormonal and anti-HER2 treatments, additional to predict the prognosis of patients with BC [12]. About 75% of BC tumors are ER and PR positive and are associated with a more favorable prognosis as they respond to endocrine therapies (estrogen receptor modulators, aromatase inhibitors, or estrogen receptor inhibitors) [13]. Tumors are interpreted as ER or PR positive when more than 1% of the tumor cell nuclei are stained (Figure 1) [14]. Approximately 20% of BC tumors have *ERBB2* gene amplification and/or HER2 protein overexpression identified by fluorescent in situ hybridization or immunohistochemistry (Figure 1), respectively. These tumors are aggressive and have a poor prognosis and response to chemotherapy [11]. Tumors that do not express ER, PR, and HER2 are referred to as triple-negative (TNBC), and they account for 10-15% of the total [15]. This subtype is a clinical challenge, as it does not respond to standard endocrine therapies, such as tamoxifen (an anti-estrogen agent against the estrogen receptor) and trastuzumab (a monoclonal antibody against HER2), and is, therefore, associated with worse prognosis and survival, increased metastases, and higher risk of recurrence compared to the other BC subtypes [15]. Hence, it is necessary to develop new effective therapies for patients with BC, especially for the subtype TNBC.

More than 15 years ago, the analysis of the gene expression profiling of BC tumors suggested a new molecular classification that divides breast carcinoma into different subtypes according to their expression: (i) luminal (expression of ER, ER regulatory genes, and/or normal luminal epithelial cells), (ii) HER2-positive (amplification and/or overexpression of the *ERBB2* gene), (iii) basal (ER, PR, and HER2-negative, but with an expression of genes from normal mammary myoepithelial and basal cells), and (iv) normal-like subtype [16,17,18]. These intrinsic subtypes differ according to the etiology of the cancer, the age of onset, and the predictive prognosis of the patient. Currently, BC characterization includes the recently mentioned immunohistochemical markers (ER, RP, and HER2), proliferation marker proteins such as Ki67, genomic markers (*BRCA1*, *BRCA2*, *PIK3CA*), immunomarkers such as PD-L1, among others [19,20].

These different subtypes and the intratumoral complexity highlight the great heterogeneity of BC, which needs to be understood in order to develop targeted therapies for each subtype. Conventional treatments for BC include surgery, hormone-blocking therapy, radiation therapy, and chemotherapy. However, 30–50% of patients diagnosed with this disease at an early stage will progress to metastatic recurrence, which may occur months or decades after the initial diagnosis despite the treatment administered [21,22].

### 1.2. Cancer Stem-like Cells in Breast Cancer

Although the largest cell burden of a tumor is formed by the so-called bulk tumor cells, a small subpopulation of cells within the tumor has recently been identified, which presents a stem cell phenotype, due to the similarities to these cells have been called “cancer stem-like cells” (CSCs) or “tumor-initiating cells” (TICs) (Figure 2) [23]. This cell population generally has the characteristic of unlimited self-renew, a hallmark of stem cells. Thus, these cells can divide symmetrically, producing two daughter cells with stem cell properties, or divide asymmetrically, producing a daughter cell with stem cell properties and a second cell that integrates with the tumor mass through differentiation mechanisms [24,25]. The symmetrical division allows excessively increased tumor growth in response to stress conditions, such as cell loss during treatments [26]. The CSCs present high tumorigenicity, several studies have proposed that they participate in all stages of cancer development, and would be responsible for the initiation, maintenance, and progression of tumors. In addition, they would participate in the expansion of the tumor to distant organs during metastasis and recurrence [27].

The lethal behavior of CSCs is mainly due to their tumorigenic capacity just described, but these cells have also been shown to be more resistant to chemotherapy, endocrine therapy, and radiotherapy compared to bulk tumor cells [28,29,30]. Accumulated evidence shows that there is an increase in the CSC ratio after conventional treatment [31]. Different mechanisms are involved in resistance to therapies, including the overexpression of membrane transporter genes of the ATP-binding cassette (ABC) family, which encode proteins responsible for pumping drugs out of the cell. These cells are more resistant to chemotherapeutic agents; have increased DNA repair; increased aldehyde dehydrogenase (ALDH) activity, and can reduce intracellular reactive oxygen species levels [32].

Induction of epithelial–mesenchymal transition (EMT) has been shown to result in the acquisition of stem cell-like properties. EMT is characterized by decreased epithelial markers such as E-cadherin and upregulation of mesenchymal proteins (Vimentin, N-cadherin, and Fibronectin). These molecular alterations cause loss of apical polarity, loss of cell-cell epithelial junctions, and promote a reorganization of the cytoskeleton which allows cancer cells to migrate, invade, and metastasize [33].

Cells exhibiting the stemness properties in breast tumors are known as “breast cancer stem-like cells” (BCSCs) [3,27,34,35]. Different BC subtypes exhibit different proportions of BCSCs, subtypes TNBC and BRCA1 hereditary are enriched for cells with the stem cells phenotype (CD44^+^/CD24^−^), while HER2^+^ tumors contain very few cells with this phenotype [36,37]. Similar results have been obtained by evaluating ALDH1, another CSC marker, finding that basal-type breast tumors are enriched for BCSC cells [38]. It has been proposed that CSCs originate from normal adult stem cells or from differentiated progenitor cells undergoing transformation. In BC, this idea is well supported, since the breast, unlike other organs, develops after birth, therefore, it requires a reservoir of adult stem cells that fulfill homeostatic and developmental functions [39].

Tumor cells and the tumor microenvironment are in a dynamic process that plays an imperative role in driving these pathways for BCSC enrichment and maintenance [40]. There are many different non-malignant cell types (such as mesenchymal stem cells, cancer-associated fibroblasts, adipocytes, endothelial cells, and immune cells) that communicate with CSCs. Besides, there is an extracellular matrix, with secreted factors, and other microenvironmental components, that are helping to induce non-CSC cancer cells to acquire a stem cell phenotype and influence the subsequent fate of these cells [40,41,42]. CSCs are plastic and dynamic and can acquire different characteristics after interactions with the tumor microenvironment [43]. In the same way, CSCs can shape the microenvironment to support both primary tumor and metastatic growth [44,45,46].

Different strategies have been used to study, isolate and/or characterize BCSCs, including (i) analysis of cell surface markers by flow cytometry; (ii) in vitro functional detection assays such as side population [47,48], the activity of aldehyde dehydrogenases ALDH1, tumorigenesis assays by soft agar colony formation, evaluation of self-renewal with sphere formation assay (mammosphere in case of breast cancer) [48,49], analysis of the expression of EMT markers and drug resistance markers (mainly ABCG1, ABCC1, and ABCG2); (iii) in vivo tumorigenesis assays by heterotopic and/or orthotopic xenograft models derived from cell lines or from patient tumors, among others [50,51].

### 1.3. Preclinical Research on Therapeutic Compounds against Biomarkers and Signaling Pathways Associated with Breast Cancer Stem Cells

The identification of biomarkers of BCSCs has had a growing interest in the BC field. Several studies have associated the expression of the biomarkers with diagnosis, therapy, and/or cancer prognosis, however, the clinical use of these CSC-specific biomarkers has been very limited. Thus, these markers could have tremendous potential not only in a better understanding of the clinical diagnosis, and tumor biology understanding but also most importantly, as new therapeutic strategies directed mainly against the BCSC population in tumors.

Currently, conventional therapeutic agents used to treat BC primarily target the tumor bulk but are less effective in killing CSCs. Therefore, it is necessary to develop better therapeutic agents focused on molecular targets expressed in BCSC, and through this strategy, target the cell population that survives conventional treatments, and is the cause of tumor recurrence, promoting the successful BC eradication. In this way, multitarget inhibitors are promising methods to overcome drug resistance and chemoresistance of BCSCs [9]. In this review, we focus on compounds that target biomarker membrane proteins of BCSCs and compounds that regulate CSC-associated signaling pathways in BC. The proteins are listed below and summarized in Table 1.

#### 1.3.1. Compounds Targeting Membrane Proteins

The increased understanding of CSC surface biomarkers has enabled significant progress in the development of CSC-targeted antibodies, and it has become an emerging technology for cancer therapy. This population is characterized by a signature of stem cell gene expression, and cell surface proteins such as CD133, CD44, EpCAM, CD49f, CD47, ABCG2, and CD24 among others that have been associated with the BCSCs phenotype (Figure 2) [52,53].

**CD44** is a receptor for hyaluronic acid and participates in the formation of complexes between extracellular components and elements of the cytoskeleton [54,55]. This receptor regulates critical cell processes including cell adhesion, migration, survival, invasion, and epithelial–mesenchymal transition through signaling pathways such as Rho GTPases, Ras-MAPK, and PI3K/AKT [54,56]. Accumulating evidence indicates that CD44 is not only a marker for CSC in BC but also in other types of cancer [57,58,59,60]. BC cells with a CD44^+^/CD24^−/low^ phenotype have a high tumorigenic capacity compared to those cells with a different phenotype that fail to form tumors [56,61]. CD44 has several isoforms as a result of alternative splicing and N- and O-glycosylation post-translational modifications. The isoform switching from CD44v to CD44s is necessary for cells to undergo EMT. Furthermore, Zhang et al., demonstrated that CD44s activate the PDGFRβ/STAT3 cascade which promotes the acquisition of BCSC characteristics. Additionally, they found that the TNBC subtype has a higher number of CSCs that mainly overexpress the CD44s isoform [58]. CD44 plays a pivotal role in CSCs in communicating with the microenvironment and regulating CSC stemness properties [62].

Hyaluronic acid (HA) has the inherent ability to target CD44 receptors, which is why it has been used to develop CSC-targeted drug delivery systems. Lapatinib (LPT) is a dual EGFR and HER2 inhibitor used as a treatment for advanced BC of the HER2 subtype, however, it has low oral bioavailability and several side effects. Agrawal et al., designed a delivery system for this inhibitor. LPT-HA-NCs are LPT nanocrystals coated with hyaluronic acid to actively target CD44 receptors on BC cells. In vitro and in vivo studies showed that LPT-HA-NCs have several cellular effects including anticancer activity of LPT-HA-NCs, increase of the intracellular concentration of LPT, promote apoptosis, and delay significantly the tumor growth [63]. Han and colleagues, developed a hyaluronan-conjugated liposome delivery system that encapsulates gemcitabine, targeting the BCSC population. This system improves the stability of gemcitabine in the bloodstream, prolonging its half-life. In vitro experiments showed a significant improvement in cytotoxicity compared to free gemcitabine. In addition, gemcitabine induces inhibition of colony-forming and cell migration. In a xenograft mouse model, a significant antitumor effect of the system was found [64]. Mitoxantrone (MTX), is an anthraquinone derivative and is used in the treatment of a wide spectrum of tumors including BC, but it has significant side effects. For this reason, Sargazi et al., developed a more effective form for administration MTX through hyaluronic acid/polyethylene glycol nanoparticles. The authors propose this delivery form as an effective nanomedicine to eliminate CD44-positive breast cancer stem cells [65]. In order to study the efficacy of nanomedicines directed against CSCs, Gener et al., developed a new model of the fluorescent label for CSCs that allows efficient detection of this subpopulation. Using this CSC reporter system, the efficacy of polymeric micelles functionalized with anti-CD44 antibodies and loaded with paclitaxel (PTX) was evaluated. The data showed that antibody-mediated targeting increases the efficacy of PTX, as well as that the mammary CSC population was sensitized to PTX treatment only when these specific micelles were used. The experiments were replicated in colon cancer cell lines, obtaining similar results [66].

**CD133** is a transmembrane glycoprotein also known as *Prominin 1.* It is often used as a biomarker for CSC enrichment in cancer, as it targets a subset of cancer cells that show a drug-resistant phenotype and an increased tumor-initiating ability in xenotransplantation assays [67]. In BC, high levels of CD133 mRNA are associated with basal, estrogen receptor-negative tumors, and higher-grade tumors (grade 3 versus 1–2) [68]. In addition, it is considered a factor of poor prognosis for metastasis-free survival in all the histological subtypes of BC analyzed to date [69]. The precise molecular mechanisms by which CD133 acts in cancer remain unclear [70].

CD133 is critical for the survival and growth of BCSCs, and antibodies against CD133 can reduce BCSC growth. Ohlfest et al., developed an immunotoxin against CD133 by fusing a gene fragment encoding the scFv portion of an anti-CD133 antibody to a gene fragment encoding deimmunized PE38KDEL. This fusion protein dCD133KDEL was shown to represent a new biological assessment tool that can be used to determine the clinical importance of eradicating CD133-positive cells since selectively inhibited CD133^+^ ductal breast carcinoma cells, and cause regression of tumor growth in mice [71]. Swaminathan et al. designed polymeric nanoparticles conjugated with an anti-CD133 monoclonal antibody (CD133NP) which were loaded with PTX (microtubule-stabilizing anticancer agent). The effect of CD133NPs on tumor-initiating cells was measured in vitro, finding that the treatment with CD133NPs significantly reduces the number of mammospheres. The anticancer effectiveness of CD133NPs in vivo in an orthotopic mouse model of BC was also investigated. This treatment effectively reduced tumor volume compared to treatment with free PTX [72]. Yin et al., designed a therapy with nanoparticles carrying anti-miR21 and the CD133 aptamer to target the stem cell marker as therapy against TNBC. Treatment with these nanoparticles had high efficacy in inhibiting tumor growth in the MDA-MB-231 xenograft model compared to the control group. Furthermore, it showed high specificity in targeting the TNBC tumor [73]. The compound 3,3′-diindolylmethane (DIM) has been studied as a chemo-preventive agent in BC. However, after the treatment with this compound, the CD133^+^/NANOG^+^ subpopulation increased, and the AKT signaling pathway was significantly activated. When using DIM in combination with the AKT inhibitor AZD5363, DIM-induced CD133 expression was decreased, further suppressing stemness and promoting anticancer activity [74].

**EpCAM** also designated CD326, is a membrane glycoprotein located mainly in the basolateral membrane of the cells, that participates in the adhesion of epithelial cells. EpCAM is recognized as one of the cell surface markers used for the identification and isolation of CSCs from various cancer types [75]. In vitro BC studies reported that the overexpression of EpCAM promotes proliferation, invasion, and metastasis in cancer cells. In addition, it has been associated with clinicopathological characteristics such as poor prognosis, larger tumor size, high histological grade, and bone metastases in BC [76]. Recent studies carried out by Dionisio et al., identified that EpCAM, along with other BCSC markers including CD44, and ALDH1, were significantly enriched in BC brain metastases compared with primary tumors. Furthermore, these patients had a worse prognosis, poor overall survival, and positive lymph nodes [77].

Catumaxomab (anti-EpCAM × anti-CD3) is a bispecific monoclonal antibody that binds to both EpCAM (on tumor cells) and CD3 (on T cells). Kubo’s group conducted an in vitro study testing the combination of catumaxomab and activated T cells in BC cell lines, finding that this combination can eliminate chemoresistant EpCAM-positive TNBC cell lines. The authors proposed that catumaxomab combined with activated T cells might act as a strong treatment to eliminate this type of cells [78]. Another immunotherapy approach developed an EpCAM aptamer-linked to small-interfering RNA chimeras to selectively knock down genes in mouse BC. This therapy was applied subcutaneously in orthotopic mouse models of HER2^+^ and aggressive TNBC tumors. The results obtained with this therapy showed a notable delay in tumor growth, and consequently, the function of tumor-infiltrating immune cells was enhanced [79].

**CD47** is an important protein in immune control as it can bind to its SIRPα ligand on macrophages, and inhibit their phagocytic capacity [80]. CD47 antibody B6H12 inhibits BCSC proliferation, asymmetric division, and expression of the mammary CSCs transcription factor KLF4 [81]. The co-expression of CD47 and HER2 markers are detected in BC patients with a poor prognosis [82]. Therefore, Candas-Green et al., generated a B6H12.2 antibody with a dual activity that blocks CD47 and HER2, this enhancing macrophage-mediated phagocytosis, and also suppressing the aggressive phenotype associated with HER2 by killing radioresistant BCSCs [82].

Although HER2 is not a marker of BCSCs by itself, different strategies directed against HER2-positive BC have been developed, including monoclonal antibodies that bind to the extracellular domain of HER2, EGFR-HER2 small-molecule kinase inhibitors, and antibody-drug conjugates. Yet, most patients with metastatic disease eventually progress to anti-HER2 therapy due to resistance to the traditional treatments [83]. Therefore, developing therapeutic strategies against the population of CSCs in HER2-positive tumors is essential. Metformin (MET) is an anti-type 2 diabetic agent effective against BC, and it has become of high therapeutical interest due to its ability to target BCSC [84]. The anticancer effect of MET in Herceptin-Conjugated Liposome (Her-LP-MET) has been evaluated in vitro and in vivo. This treatment produced greater inhibition of BCSC proliferation in vitro compared to free MET. The anti-migratory effect of Her-LP-MET on BCSC was enhanced when used in conjunction with doxorubicin (DOX). In the same way, in a mouse model, Her-LP-MET combined with free DOX was more effective, reducing tumor mass and prolonging tumor remission. This Her-LP-MET formulation is proposed as a new therapy to efficiently targets BCSCs [85]. Other therapeutic strategies have also been tested for cells that express the HER2 receptor. The combined treatment of a carbon-ion beam and LPT, effectively destroys the proportion of HER2-positive BC stem cells (ESA^+^/CD24^−^), decreasing cell viability, the formation of spheroids, and promoting apoptosis [86].

#### 1.3.2. Compounds That Regulate Signaling Pathways

In recent years, various strategies have been proposed to eliminate BCSCs. These include the blockade of signaling pathways associated with the acquisition of stem cell characteristics [87]. BCSCs exhibit dysregulation of critical signaling pathways such as Notch [88], Hedgehog (Hh) [89], transforming growth factor-β (TGF-β) [90], Wnt/β-catenin [27,91], STAT3 [92,93], PI3K/AKT/FOXO [94], and NF-κB [95]. These cellular signalings allow BCSCs to have a greater capacity to proliferate and tolerate hostile environments. BCSCs also express metabolic markers of stem cells, such as aldehyde dehydrogenase (ALDH1) [96], transcription factors such as OCT4 (octamer-binding factor 4), SOX2 (SRY-box transcription factor 2), NANOG, KLF4, and MYC, that act as key regulators of pluripotency and self-renewal [97] (Figure 2). Importantly, these pathways also play a fundamental role in the normal development of the mammary glands.

**The Wnt/β-catenin pathway** is a critical pathway in embryonic development and tissue homeostasis. In cancer, it plays a fundamental role in the functioning of the CSC, orchestrating self-renewal and cell differentiation [98]. In BC, more than half of the tumors have this active pathway, which is associated with decreased patient survival of these patients. Given the importance of this signaling pathway in cancer, a large number of inhibitory compounds targeting proteins of this pathway have been designed [99]. The group of Jang et al., designed a selective small-molecule inhibitor called CWP232228 (US Patent 8,101,751 B2), which prevents β-catenin from binding to TCF in the nucleus. In vitro and in vivo studies demonstrated that treatment with CWP232228 targets BCSC populations, blocking the formation of secondary spheres in cells resistant to conventional chemotherapy. Besides, the inhibitor suppresses tumor formation in a murine xenograft model and metastasis by inhibiting the growth of both, bulk tumor cells and BCSCs. It is proposed that this inhibitor would have significant therapeutic potential for the treatment of BC [100].

**The TGF-β signaling pathway** has biological activity capable of inducing the oncogenic transformation of non-cancerous cells. TGF-β promotes cell survival and proliferation, in addition to stimulates the maintenance of stemness in the BCSC population [101]. Several studies suggest that TGF-β-induced CSC accumulation in TNBC is a mechanism of drug resistance in TNBC [37]. Park et al., reported that blockade of TGF-β signaling with the inhibitor EW-7197, in combination with PTX, reduce the EMT process and BCSC population (induced by PTX treatment) by suppressing Snail expression. Besides, a reduction of mammosphere formation efficiency, a decrease of ALDH activity and CD44^+^/CD24^−^ phenotype, and diminish of pluripotency regulators (OCT4, NANOG, KLF4, MYC, and SOX2) was observed. This combined treatment enhances the therapeutic effect of PTX by decreasing lung metastasis, and increasing survival time in vivo [102]. Another study also shows that PTX increases the BCSC population through TGF-β signaling. The inhibition of this pathway with the inhibitor LY2157299 reduced the population of BCSCs resistant to PTX in vivo, and abolished the tumor-initiating potential of BCSCs after chemotherapy [103].

**In the Notch signaling pathway**, Notch is a ligand-activated transmembrane receptor (delta-like (DLS) and Jagged), which undergoes serial cleavage by γ-secretases, leading to the intracellular portion of Notch translocating to the nucleus where activates the expression of downstream transcription factors [104]. It has been reported that this pathway is over-activated in BC, and would be promoting chemo/radioresistance, both in BC cells and in BCSC [105]. Several research groups hypothesize that Notch inhibition will allow BCSC to be eliminated, controlling the progression of BC [106]. Currently, there are three main clinical methods used to inhibit Notch signaling. These include γ-secretase inhibition (GSI), antibodies against Notch receptor or ligand antibodies, and a combination therapy with other pathways [107]. A study conducted by Li et al., evaluated the in vitro and in vivo effect of five inhibitors directed to BCSC in TNBC. The tested inhibitors were DAPT, GDC-0449, Salinomycin, Ruxolitinib, and Stattic, that target Notch (γ-secretase), Hedgehog (SMO), Wnt (β-catenin), JAK/STAT, and JAK (STAT3) pathways respectively. The inhibitors were found to have antiproliferative and proapoptotic functions, along with suppressing invasion and decreased self-renewal (markedly diminished size and number of mammospheres) in BCSCs, in TNBC cells. Also, these inhibitors reduced the expression or phosphorylation of their downstream signaling target molecules in a dose-dependent manner. Furthermore, these five inhibitors suppress the tumor-forming ability of the TNBC stem cell line HCC1806 when pretreated with the inhibitors in xenograft mouse models [15]. In another study, mesoporous silica nanoparticles (MSNPs) have been designed as vehicles for the targeted delivery of GSI to block Notch signaling. In vivo analysis, showed that MSNP-GSI nanoparticles enhanced Notch signaling inhibition compared to the free drug. These data suggest that MSNP-GSI is an attractive platform for the targeted delivery of anticancer drugs and, specifically, against the population of CSCs [108].

The FK506 binding protein (FKBPL) is considered a prognostic and predictive biomarker of BC since it has antitumor and antiangiogenic activity. FKBPL has been shown to have the ability to target CD44-positive cells and reducing their population. Also, FK506-derived peptides (ALM201 and AD-01) inhibit the BCSC resistant to endocrine therapy. In addition, downregulation of DLL4 and Notch4 has been reported to reduce migration, invasion, and pulmonary metastasis of BC [109,110].

**The Hedgehog (Hh) signaling pathway** includes three secreted ligands, of which Sonic Hedgehog (SHH) is the most widely expressed, followed by transmembrane receptor/coreceptor Patched (PTCH) and Smoothened (SMO). Activation of this pathway occurs when the Hh ligands bind to the PTCH transmembrane receptor, which regulates the SMO transmembrane protein The binding induces the activation of the GLI oncogene, which is a transcriptional effector of the Hh pathway [111]. This pathway regulates the maintenance, self-renewal, survival, and proliferation of BCSC [37,112]. In TNBC, Hh signaling has been associated with high-grade, highly proliferative cancer, increased metastasis, and poorer disease-free survival [113]. GANT61 is a non-canonical Hh pathway inhibitor. GANT61 was able to decrease E2-induced GLI1/2 activation accompanied by a decrease in the proportion of CSCs in ER-positive BC cells [89]. Another study evaluated the effect of GANT61 on TNBC cells and found that this inhibitor significantly decreased the proportion of BCSCs in all TNBC cell lines analyzed. The combined treatments of GANT61 and PTX greatly enhanced anti-cell growth and/or anti-CSC activities. These results suggest that GANT61 has the potential as a therapeutic agent in the treatment of patients with TNBC [114].

**In the STAT3 signaling pathway,** the tyrosine kinase Janus-activated kinase 2 (JAK2) is hyperactive in HER2-positive and TNBC tumors [115]. JAK2 phosphorylates and activates STAT3 to induce its nuclear translocation, increasing the expression of genes that promote CSC turnover [116]. Dysregulation of the JAK2-STAT3 pathway is associated with poor clinical outcomes, and it has been investigated as a possible therapeutic target for BCSC [92,93]. The effect of combined inhibition of the JAK2-STAT3 (by Ruxolitinib and Pacritinib) and SMO-GLI1/tGLI1 (by Vismodegib and Sonidegib) signaling pathways on the BCSC population has been evaluated. The combined treatment with these inhibitors suppressed the high CD44^high^/CD24^low^ BC stem cell population compared with vehicle or either every agent alone. Also, the simultaneous inhibition of the JAK2-STAT3 and SMO signaling pathways suppresses the orthotopic growth of TNBC tumors and reduces metastasis in vivo [93]. Tamoxifen (TAM) is the first-line treatment for ER^+^ receptor-positive BC. However, resistance to this drug is the main obstacle in clinical practice. Therefore, new therapies are needed to treat TAM resistance. Liu et al., evaluated the effect of Napabucasin, a small STAT3 inhibitor, and found that it attenuates BC cell resistance to TAM by reducing the population of BCSCs, which was evidenced by the decrease in stem markers (OCT4, NANOG, and SOX2), in combination with a reduction in the ability to form spheroids and a decrease of the ALDH1 activity [91].

**Aldehyde dehydrogenase-1 (ALDH1)** is a detoxification enzyme that catalyzes the oxidation of intracellular aldehyde substrates. Often, ALDH1 is used to isolate and identify normal cell populations enriched in stem and progenitor cells, as well as BCSCs in cancerous tissues [96]. In BC, ALDH1 positivity has been correlated with high histological-grade tumors, HER2 overexpression, and the absence of the expression of estrogen and progesterone receptors [117,118]. In addition, it was reported that the expression of this marker is related to a poor prognosis [118]. ALDH1^+^ BCSCs are enriched in basal-like and HER2-overexpressed tumors [38]. Furthermore, the expression of ALDH1 contributes to both chemotherapy and radiation resistance [119].

Analyses of gene expression profiles have shown that there are two different subpopulations of BCSCs, those that display a CD44^+^/CD24^−^ phenotype, which is a mesenchymal marker, located in the ductal structures frequently associated with ductal branch points [120]. Conversely, ALDH^+^ BCSCs, are epithelial and highly proliferative cells located in an abluminal location within lobules. These two BCSC subgroups can transit through these two dynamic states [120,121,122]. Indeed, BCSC CD44^+^/CD24^−/low^/ALDH1^+^ phenotype is a very small population within the tumor [38]. This population is more tumorigenic, and it is able to recapitulate the tumor the after reduction of the cell population sensitive to therapy, which leads to relapse of the disease [123]. Disulfiram (DSF) is a drug used as a treatment for alcoholism that produces an irreversible inhibition of ALDH [124]. Its effect on the CSC population in BC was evaluated and it was found that DSF eradicates BCSC, inhibits CSC marker expression such as SOX2 and OCT4, and reverses PTX and cisplatin resistance in MDA-MB-231 PAC10 cells (resistant to cytotoxicity) [125]. A subsequent study investigated the mechanism of action of DSF on BCSCs. They found that DSF eliminated BCSCs and inhibits tumor development in vivo through suppression of HER2/AKT signaling. [126]. Another study showed, for the first time, that DSF suppresses stem properties in TNBC by targeting the STAT3 signaling pathway [127].

### 1.4. Drug Repurposing to Target Breast Cancer Stem-like Cells

New drug development is a significantly expensive multi-step process, involving drug design and synthesis, as well as re-testing in animal models, to ensure precise safety and efficacy. An alternative strategy to *de novo* drug development is drug repurposing, which uses molecules already approved by the FDA for new therapeutic indications that can effectively eradicate BCSCs and control metastatic disease in BC. The latter represents an effective strategy to rapidly improve the prognosis of BC patients [128]. Salinomycin and pyrvinium pamoate are 2 FDA-approved drugs that have been redirected to the treatment of BCSC [129].

Salinomycin (SLM) is an antibiotic identified as an effective anti-BCSC compound. SLM reduces the proportion of CSC by >100-fold compared to PTX and inhibits mammary tumor growth in vivo. Besides, it decreases BCSC-associated gene expression through modulation of WNT and Hh signaling pathways [130,131,132]. SLM has also been tested in conjunction with LBH589 (Panobinostat, a histone deacetylase inhibitor) as a new therapy against TNBC. The combined treatment of these drugs showed effective and synergistic inhibition of tumor growth of an ALDH1^+^ TNBC xenograft mouse model, by inducing apoptosis, cell cycle arrest, and EMT regulation, with no apparent associated severe toxicity. Therefore, this drug combination could offer a novel therapeutic strategy for patients with TNBC [133]. A recent study demonstrated that SLM and its C20-propargylamine derivative (Ironomycin 2) eliminate BCSCs by ROS production, induced after iron accumulation in lysosomes [134]. Another delivery system for SLM was designed by Muntimadugu et al., They prepared nanoparticles loaded with SLM or PTX and coated them with hyaluronic acid to target BCSC cells that overexpress the CD44 receptor. The results obtained in this study showed a co-eradication of CD44^+^ BCSCs as well as bulk tumor cells. Combination therapy of HA-coated SLM nanoparticles and PTX nanoparticles is a promising approach to overcome cancer recurrence due to resistant BCSCs [135].

Pyrvinium pamoate (PP) is an anthelmintic drug and a suppressor of the WNT pathway. Xu’s group evaluated the effect of the pharmacological blockade of this pathway with PP on the BCSC population as a possible therapy to treat BC. They found that this drug inhibits the in vitro proliferation and self-renewal of BCSCs in BC cell lines. Moreover, PP decreases the content of CD44^+^/CD24^−/low^ and ALDH^+^ BCSCs in a panel of BC cell lines. In a xenograft model of BC cells, it was shown that cells pretreated with PP strongly delayed tumor development. Furthermore, an in-depth analysis revealed that PP inhibits the activity of the WNT pathway and the expression of stem regulators (NANOG, SOX2, and OCT4) [136]. A related study identified the metabolic consequences of PP. This drug inhibits the anabolic metabolism of fatty acids and cholesterol, both are essential for the survival of BCSC in TNBC [129].

Cui et al. screened a drug library of FDA-approved compounds (Prestwick Library) to identify inhibitors of mammary CSCs. They found that the treatment with benztropine mesylate decreases the cell subpopulation that shows both ALDH1 activity and CD44^+^/CD24^−^ phenotype, in addition to the suppression in the formation of mammosphere and the decrease of the self-renewal properties of BCSC. Additionally, in vivo studies showed that benztropine mesylate inhibited the potential for tumor initiation of the 4T1 cell line in a mouse model. Their findings show that benztropine mesylate could be a good BCSC inhibitor both in vitro and in vivo [137].

**Table 1 cells-12-00720-t001:** Preclinical studies of therapeutic compounds directed against breast cancer stem-like cells.

Therapeutic Molecule	Targeting	Effects	Ref
LPT-HA-NCs	CD44	Anticancer activity and delays tumor growth	[63]
Hyaluronan-conjugated liposomes encapsulating gemcitabine	CD44	Inhibits migration and colony formation ability in vitro assays. Potent antitumor effect in vivo	[64]
Polymeric micelles functionalized with anti-CD44 antibodies and loaded with paclitaxel	CD44	Increases the sensitivity of mammary CSCs to PTX treatment	[66]
FKBPL and its peptide derivatives	CD44	It inhibits tumor growth and CD44-dependent antiangiogenic activity. It also targeting CD44-positive BCSCs, it also inhibits migration, invasion, and formation of mammospheres resistant to endocrine therapy. It reduces lung metastases in an in vivo model by downregulating DLL4 and Notch4	[109,110]
Fusion protein dCD133KDEL	CD133	Selectively inhibits CD133^+^ ductal breast carcinoma cells, causing regression of tumor growth in mice	[71]
Polymeric nanoparticles conjugated with an anti-CD133 antibody loaded with paclitaxel	CD133	Reduces the number of mammospheres and cell colonies. In animal models decreases tumor volume	[72]
3WJ/CD133_apt_/anti-miR21 nanoparticles	CD133	High specificity in targeting TNBC tumor. High efficacy in tumor growth inhibition	[73]
Catumaxomab	EpCAM	Eliminates chemoresistant EpCAM-positive triple-negative cell lines	[78]
EpCAM aptamer-linked small-interfering RNA chimeras	EpCAM	Delays tumor growth and improves function of tumor-infiltrating immune cells	[79]
B6H12	CD47	Inhibits proliferation, asymmetric division, and KLF4 expression in BCSC	[81]
B6H12.2	CD47 and HER2	Inhibits the phagocytosis capacity of macrophages and abolishes the aggressive BCSCs phenotype of BCSCs	[82]
Metformin in Herceptin-Conjugated Liposome	HER2	Inhibits proliferation and migration of BCSCs in vitro. In an animal model, reduces tumor mass and extends tumor remission	[85]
Carbon-ion beam combined with lapatinib	HER2	Decreases BCSC ratio, cell viability, spheroid formation, and promotes apoptosis	[86]
CWP232228	Wnt/β- catenin pathway	Inhibits sphere formation, decreases the proportion of BCSCs resistant to conventional therapy, and further reduces tumor growth and metastasis in vivo	[100]
EW-7197	TGF-β signaling pathway inhibitor	In combination with PTX reduces the EMT process and proportion of BCSCs	[102]
LY2157299	TGF-β signaling pathway inhibitor	Reduces the population of BCSCs resistant to PTX and abolishes the tumor-initiating potential of BCSCs after chemotherapy	[103]
Napabucasin	STAT3 inhibitor	Attenuates the BC cell resistance to tamoxifen by reducing the BCSC population	[91]
DAPT, GDC-0449, Salinomycin, Ruxolitinib, and Stattic	Notch, Hedgehog, Wnt/β-catenin, JAK, and JAK/STAT3, respectively	Antiproliferative and proapoptotic activities, suppress invasion and self-renewal of BCSC in vitro. They suppress the ability to form tumors in vivo	[15]
GANT61	Non-canonical hedgehog pathway	Decreases E2-induced GLI1/2 activation and the proportion of BCSCs in ER-positive BC and TNBC cells	[89,114]
Disulfiram	STAT3 pathway	Inhibits STAT3 pathway suppressing stem properties in BC	[125,127]
Salinomycin	WNT and Hh signaling pathways	Reduces the proportion of BCSC and inhibits mammary tumor growth	[134]
Pyrvinium pamoate	Selective WNT pathway inhibitor	Inhibits proliferation and self-renewal of CSCs in BC cell lines. In the xenograft model of BC, strongly delays tumor development	[136]
Pranlukast	CD49f	Reduces the CSC population in TNBC cells	[138]
Benztropine Mesylate	NS	Decreases cell subpopulation with ALDH1 activity and CD44^+^/CD24^−^ phenotype, diminishes mammosphere formation, and BCSC self-renewal capacities	[137]

BC: breast cancer; BCSC: breast cancer stem cell; TNBC: triple-negative breast cancer; GLI: glioma-associated oncogene; PTX: Paclitaxel; HER2: Human epidermal growth factor receptor 2; GSI: γ-secretase inhibitor; NS: not specified.

CD49f (α6 integrin; ITGA6) is expressed on breast CSCs and functions in the maintenance of stemness. Pranlukast, a drug used to treat asthma, works as an antagonist of CD49f. Pranlukast treatment decreased BC cell clonogenicity in the mammosphere formation assay, and also diminished CD44 and SOX2 expression, and tumorigenicity in vivo, showing that this drug reduces the CSC population in TNBC cells [138].

Wedelolactone is a promising anticancer drug [139]. To enhance its biological activity, wedelolactone-encapsulated PLGA nanoparticles (nWdl) were formulated. In vitro assays showed that nWdl delayed migration, invasion, and EMT in MDA-MB-231 cells compared to the free drug. Wedelolactone significantly upregulates mesenchymal markers, such as N-cadherin, Vimentin, Twist, Snail, and Slug. Moreover, it also upregulated epithelial markers E-cadherin and cytokeratin-19. Regarding the population of BCSCs, it was observed that the expression of ALDH^+^ cells was significantly decreased when treated with nWdl combined with PTX relative to PTX alone. Also, nWdl significantly inhibited the formation of mammospheroids, and reduced the expression of pluripotency and chemoresistance markers such as ABCG2, SOX2, and ALDH1. In vivo studies showed that nWdl effectively reduced tumor volume and BCSC population (CD44^+^/CD24^−^/^low^) in xenograft models [140].

One of the main challenges in BCSC research is at the translational level, moving from laboratory experiments toward appropriate clinical trials to test new compounds targeting BCSCs. The challenge includes to assessing their efficacy in killing the BCSC population of patient tumor cells [141].

### 1.5. Breast Cancer Stem-like Cell Targeted Therapies in Clinical Trials

In the last decade, a wide variety of treatments have been developed in clinical trials based on the classification of BC molecular subtypes. Current therapies for BC (chemotherapy and hormone therapy) are effective in killing cancer cells and controlling tumor growth. Most anticancer compounds only affect tumor cells that are in a proliferative state, and they leave behind a small population of dormant CSCs. These cells exhibit increased tumorigenic potential, and often acquire an EMT phenotype, leading to subsequent relapse and therapy-resistant metastases [142]. It has been reported that almost all patients with metastatic BC, and a quarter of those with early BC will relapse despite the initial response [106]. Nevertheless, only a few clinical trials evaluate the effectiveness of conventional treatments against this specific population of cancer cells, even when targeting BCSCs seems to lead to promising therapies [107].

Antitumor therapies specifically targeting BCSCs have long been advised to be administered in conjunction with the traditional chemotherapeutic regimen with the goal of preventing relapse. Interestingly, there are several clinical trials being developed to determine the efficacy of specific therapies against the BCSC population [143]. In addition, these clinical trials have helped to understand the biology, and regulatory mechanisms of BCSCs, and how they respond to new therapies.

To identify new compounds or treatments that might be evaluated in a clinical trial setting against the BCSCs population, a search on the website ClinicalTrials.gov was performed using the keywords “breast cancer stem cells”. Almost 150 clinical trials were found to date, but only a small part of them were actually therapies directed toward BCSCs. The vast majority were peripheral stem cell transplants as a treatment for patients with BC. Clinical trials targeting BCSCs will be detailed below and are summarized in Table 2.

#### 1.5.1. Antibodies Targeting Breast Cancer Stem-like Cells

One of the clinical trials with therapeutic agents targeting CSC in BC is the AVASTEM NCT01190345 trial, which evaluated the anticancer capacity of preoperative Bevacizumab, a monoclonal antibody targeting the vascular endothelial growth factor (VEGF) receptor, along with conventional therapy in 75 participants with BC. The proportion of BCSCs was measured by the number of cells positive for the ALDH1 marker after 4 cycles of treatment. This trial does not confirm the impact of Bevacizumab on breast CSC cells, as it did not change BCSC rates compared to standard neoadjuvant chemotherapy [144]. Therefore, this antibody still has a controversial role in the treatment of BC [145].

In the clinical trial, NCT01424865, the expression of the cancer stem cell biomarker CD44^+^/CD24^−^ y ALDH1 was evaluated as a predictor of response to Trastuzumab in 1874 samples from BC patients previously treated in the NSABP-B-31 trial. The results of this clinical trial suggest that the CD44^+^/CD24^−^ phenotype may be used as a predictor of clinical outcome, and as a predictor of response to Trastuzumab treatment in patients with HER2-positive primary BC [146].

Schmidt et al. conducted a randomized phase II study to investigate the efficacy of adecatumumab (anti-EpCAM) as monotherapy in 117 patients with metastatic BC. They found that the probability of tumor progression was significantly lower in patients who received high doses of adecatumumab, and who expressed high levels of EpCAM. But the use of this antibody in BC requires more research [147]. Subsequently, the same group evaluated the combined effect of adecatumumab with Docetaxel (DTX) in a phase IB trial in 31 women with advanced-stage BC, especially for high EpCAM-expressing tumors. They declare that this combination therapy is safe, tolerable, and potentially active in advanced-stage BC [148]. Another clinical trial directed at EpCAM is NCT02915445, in which the safety of EpCAM CAR-T cells was determined in 30 patients with nasopharyngeal carcinoma or BC expressing high levels of EpCAM. Chimeric antigen receptor-modified T cells (CAR-T) have the ability to target tumor antigens such as EpCAM, and can specifically recognize, bind to, and kill antigen-positive tumor cells [149]. In these patients, the potential to inhibit tumor progression will be measured. The results of this clinical trial are expected to provide a new treatment strategy for patients with BC, but nevertheless, the results of this study are not yet known.

The clinical trial NCT02254005 evaluated the maximum tolerable dose, safety, pharmacokinetics, and efficacy of a single dose of the Bivatuzumab mertansine antibody in female patients with CD44v6-positive metastatic BC. The disease was stabilized in 50% of the treated BC patients regardless of the dose level [150].

The currently running clinical study NCT02776917 is evaluating the combined treatment of Cirmtuzumab, a monoclonal antibody directed against the ROR1 (receptor-tyrosine-kinase like orphan receptor 1), with PTX, in 22 patients with unresectable BC, metastatic or locally advanced. After 4 weeks of treatment, ROR1 expression levels will be measured and the proportion of CSCs will be evaluated through ALDH1 and CD133 markers. Additionally, primary tumor samples will be compared before and after treatment.

#### 1.5.2. Inhibitors of Breast Cancer Stem-like Cells Signaling Pathways

Several research groups have hypothesized that inhibition of the Notch pathway is effective to eliminate BCSCs, and thus, controlling advanced BC [106]. In a first phase I clinical trial (NCT00106145) conducted in 103 patients with metastatic or advanced BC, the safety/tolerability and efficacy of the Notch signaling pathway inhibitor, MK-0752, were determined. The study shows a clinical benefit of the γ-secretase inhibition, which led to further trials in combination with other therapies to maximize the clinical benefit of the strategy [151]. Schott et al., suggest that the combination of cytotoxic chemotherapy together with MK-0752 would give better results to efficiently eliminate BCSC, and control the disease. Therefore, preclinical and clinical studies were carried out in parallel [106]. Using human breast tumor graft studies, they evaluated the impact of this inhibitor on the BCSC population, and the efficacy of a combination treatment of MK-0752 with DTX. The study demonstrates that MK-0752 targets the BCSC population. In parallel, a clinical trial (NCT00645333) was carried out in 30 patients with advanced BC, treated with increasing doses of MK-0752 together with DTX. This work was designed to determine the maximum tolerated dose of the inhibitor, administered sequentially with DTX, and to evaluate BCSC biomarkers in tumor biopsies. A decrease in CD44^+^/ CD24^−^, and ALDH^+^ was demonstrated in tumors from patients with this combination therapy [106]. The effect of MK-0752 in combination with TAM or Letrozole to treat early-stage BC patients was also tested in the clinical trial NCT00756717, but its effect on BCSC markers was not evaluated.

Another γ-secretase inhibitor is RO4929097, in preclinical studies, this inhibitor showed a significant reduction in anchorage-independent growth and sensitized BC cells to ionizing radiation, both characteristics of CSCs [152]. Additionally, its antitumor activity in patients with advanced, metastatic, or recurrent triple-negative invasive BC has been studied in clinical trials (NCT01151449). The effect of RO4929097 in combination with other drugs like PTX and Carboplatin in TNBC [153] or in combination with Exemestane in patients with advanced or metastatic BC (clinical trial NCT01149356 [154], has been also evaluated, but its anti-BCSC effect has not yet been addressed.

PF-03084014 is another selective, reversible, non-competitive γ-secretase inhibitor that blocks the Notch signaling pathway. In the phase I trial NCT01876251, they evaluated the maximum tolerated dose, safety, pharmacokinetics, and antitumor activity of PF-03084014 in combination with DTX, in 30 patients with advanced TNBC [155]. Subsequently, in a phase II trial NCT02299635, the effect of PF-03084014 was evaluated in patients with advanced triple-negative breast cancer with or without genomic alterations in Notch receptors, however, its specific anti-CSC activity was not evaluated. Zhang et al., studied the antitumor efficacy of PF-03084014 alone and in combination with DTX against TNBC *in vitro*. They found that PF-03084014 enhanced the antitumor efficacy of DTX in TNBC xenograft models, through impairment of the Notch Pathway. Furthermore, PF-03084014 significantly reduced tumor-initiating cells or CSCs in the BC xenograft model [156].

Reparixin is a small-molecule inhibitor of the CXCR1 receptor, identified as an investigational drug targeting CSCs. It has been reported that this inhibitor selectively reduced the BCSC population in BC cell lines, and also was able to specifically target the CSC population in human BC xenografts. The latter delays tumor growth rate, and reduces metastasis [157]. A pilot study (NCT01861054) evaluated the effect of reparixin (administered orally) on BCSCs in primary tumors in a population of 20 patients with early BC. BCSCs were measured in tissue samples using CD44, C24, and ALDH1 biomarkers, as well as EMT markers (Snail, Twist, and Notch), among other proteins. ALDH^+^ and CD44^+^/CD24^−^ markers measured by flow cytometry decreased by >20%. However, these results could not be confirmed by immunofluorescence due to the very low number of BCSCs [158]. Subsequently, this same group conducted a phase 2 FRIDA NCT02370238 trial where they evaluated progression-free survival in 123 patients with triple-negative metastatic BC treated with PTX in combination with reparixin. In this study, the BCSCs positive for CD44^+^/CD24^−^, and ALDH^+^ markers were measured. However, the combined treatment did not improve progression-free survival in these patients [159].

LGK974 is an inhibitor of Wnt signaling [160]. The phase I clinical trial (NCT01351103) will evaluate its effectiveness and safe administration as a single agent, and in combination with PDR001 (anti-PD-1 immunotherapy) in adult patients with solid malignancies, including TNBC. The results would support the promising use of Wnt inhibitors as candidates to target BCSCs. Previous studies found that this inhibitor can influence the recruitment of immune cells to tumors and can enhance checkpoint inhibitor activity [161].

Preclinical studies together with a phase I/II clinical trial (NCT01118975) determined the effects of the combination of vorinostat (inhibitor of histone deacetylase, which induces epigenetic changes) and LPT (a dual EGFR and HER2 inhibitor) in BCSCs population. In these studies, the combination of vorinostat and LPT was shown to reduce the BCSC population, measured through the surface markers CD49f and CD44^+^/CD24^−/low^. Additionally, the ALDH1 activity was evaluated in three BC cell lines. A reduction in the formation of mammospheres, decreased self-renewal proteins (BMI-1 and β-catenin), diminished EMT markers (Twist1 and Vimentin), and the inhibition of cell migration were also observed. The phase I/II clinical trial conducted in patients with BC positive for advanced HER2 demonstrated that the combined treatment of vorinostat and LPT is feasible and safe, with controllable side effects. Interestingly, patients who continued on vorinostat and LPT did not develop any new metastatic sites, supporting the preclinical results that show that this combination targets the BCSC population and prevents metastases. Additional studies are needed to further validate these results [162]. Another phase II clinical trial with LPT is NCT01868503 in which the combination of LPT ditosylate and radiation therapy work in treating patients with locally advanced or locally recurrent BC was tested. This study evaluated the change in BCSC proportion, viability by flow cytometry, and gene expression profiling after combined therapy. The results of this trial are not yet reported. The clinical trial NCT00524303 conducted in 100 women with invasive BC overexpressing HER2, evaluated the effect of LPT in combination with standard neoadjuvant chemotherapy (5FU, Epirubicin, Cyclophosphamide, and PTX). However, the data obtained from BCSCs were of poor quality, therefore, they could not be analyzed.

Ruxolitinib is a recently discovered drug that has been shown to block the IL6/JAK/STAT3 pathway by targeting JAK1 and JAK2 [163]. This inhibitor is being tested in a phase II clinical trial (NCT02876302) as a possible treatment for inflammatory breast cancer in 23 patients in combination with PTX. The distribution of the CD44^+^/CD24^−^ stem cell population and the inhibition of JAK and pSTAT3 expression in BC tumors before and after treatment with ruxolotinib will be evaluated. This trial has not yet finished.

The retrospective study S9313B (NCT00949013) included 1600 tumor tissue samples recruited from women with early-stage BC. In this trial, the expression of the BCSC marker, ALDH1, was evaluated as a predictor of response to adjuvant chemotherapy (DOX and Cyclophosphamide), together with other biomarkers such as HER2, ER, and PR, but the data has not been published.

## 2. Conclusions

The great heterogeneity and complexity of BC represents a challenge in the search for new treatments that are effective in reducing the risk of metastasis and recurrence. The accumulated evidence indicates that the current subtype classification of breast cancer tumors is not sufficient to choose appropriate treatments. Instead, the representation and behavior of existing BCSCs in the tumor mass need to be also considered. This finding suggests the need to include the analysis of a variety of specific markers expressed throughout the natural history of the tumor, and in response to conventional treatments in this cell population. This evaluation will allow the development of treatments that combine targeting the inhibition of the mechanisms associated with the proliferation and survival of BCSCs.

As described here, there are several preclinical studies evaluating new therapeutic alternatives directed toward BCSCs. The strategies that have shown the most promising results not only consider the use of inhibitory compounds directed at the most representative signaling pathways of this type of cell (BCSCs), but also consider the use of innovative nanoparticles, and explore targeted delivery by coating these formulations with molecules that recognize surface markers expressed by BCSCs. These approaches have shown better effectiveness and specificity in the treatment of chemoresistant breast cancer than the use of these compounds as free formulations or standard treatments or conventional therapies alone. In order to obtain more comprehensive evidence to guide the design of clinical studies evaluating the effectiveness of treatments for BC patients, it is essential to use not only mammosphere models, but also patient-derived xenograft (PDX) models that accurately recapitulate the tumor microenvironment, including the presence of BCSCs in the tumor. Many of the clinical studies primarily evaluate the effectiveness of treatments targeting this type of cells alone, or in combination with standard treatments. However, the variability in the results obtained highlights the need to improve the translation of findings from preclinical studies to their evaluation in patients. Therefore, the available evidence suggests that the most promising treatments should consider the different subtypes of BC, the expression of specific BCSC markers, the use of nanoparticles targeted toward these markers, and the combination of these new treatments with the most successful current standard treatments or conventional therapies.

## Figures and Tables

**Figure 1 cells-12-00720-f001:**
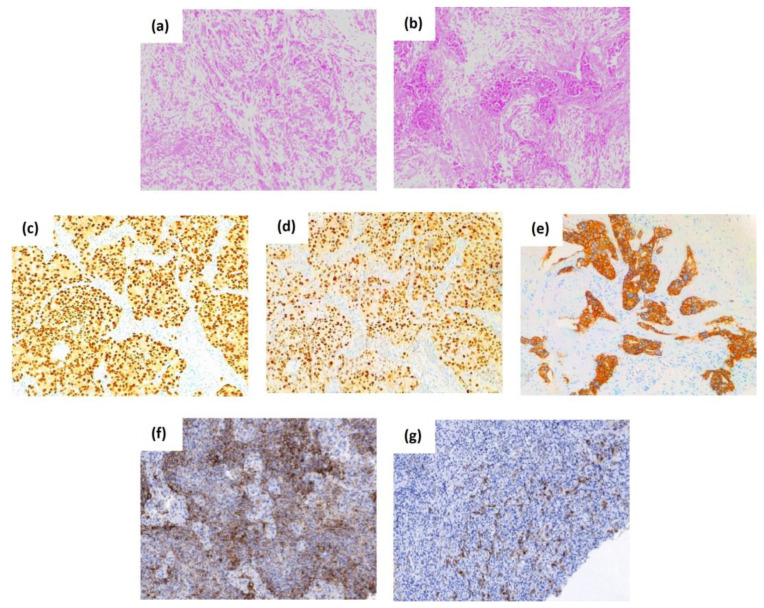
The histological subtypes, common and stem cell-associated markers of breast cancer. The top panel shows representative histological micrographs stained with hematoxylin-eosin of lobular-type (**a**) and ductal-type breast carcinoma (**b**). Representative images of the expression of the first immunohistochemical biomarkers that have laid the foundations for the classification of the BC subtypes, the estrogen receptor (**c**), the progesterone receptor (**d**), and the human epidermal growth factor receptor 2 or HER2 (**e**). The bottom panel shows the expression of the breast cancer stem cell-associated biomarkers CD44 (**f**), and CD133 (**g**). Original magnification 100×.

**Figure 2 cells-12-00720-f002:**
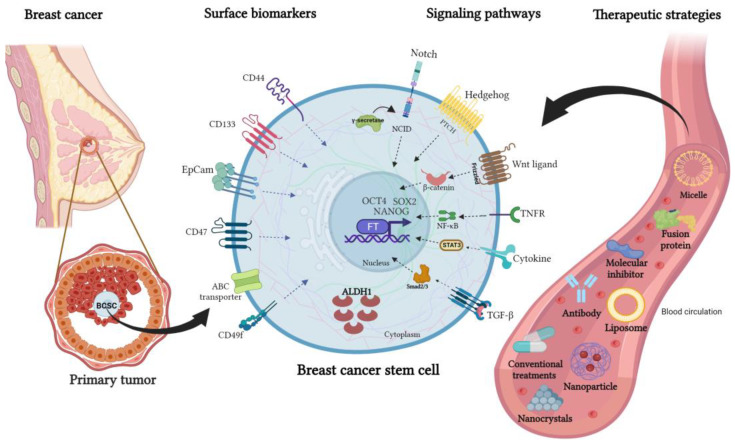
Surface biomarkers, signaling pathways, and an overview of new therapeutic strategies to target breast cancer stem cells. Mammary CSCs express cell surface proteins that are used as biomarkers such as CD133, CD44, EpCAM, CD47, CD49f, and ATP-binding cassette (ABC) transporters. In addition, this cell subpopulation has active signaling pathways, being the main ones Notch, Hedgehog, TGF-β, Wnt/β-catenin, STAT3, PI3K/AKT, and NF-κB. The expression of transcription factors (Nanog, SOX2, and OCT4) allows them to acquire and maintain the characteristics of self-renewal, pluripotentiality, tumorigenic capacity, resistance to therapies, and high metastatic capacity. Several therapeutic strategies have been developed to deliver BCSC-targeted drugs, such as liposomes, micelles, nanoparticles, nanocrystals, and antibodies directed against membrane proteins or inhibitors of BCSC-associated signaling pathways.

**Table 2 cells-12-00720-t002:** Clinical trials of therapeutic agents targeting breast cancer stem-like cells.

NCT Number	Clinical Trial	Drug Name	BCSCs Biomarker Evaluated	Sample Size	Status
NCT01190345	Anticancer Stem Cell Activity of Preoperative Bevacizumab and Chemotherapy in Breast Cancer (AVASTEM)	Bevacizumab is mAb anti-VEGF	In tissue samples, ALDH1 activity was measured	75 patients with BC	Phase II, completed
NCT01424865	Cancer Stem Cell Biomarkers as a Predictor of Response to Trastuzumab in Samples From Patients With Breast Cancer Previously Treated in the NSABP-B-31 Trial	Trastuzumab is an mAb anti-HER2	In tissue samples, ALDH1 activity was measured	1874 samples from patients with breast cancer	Unknown
NCT01973309	A Study of Vantictumab (OMP-18R5) in Combination With Paclitaxel in Locally Recurrent or Metastatic Breast Cancer	Vantictumab is a mAb binds to frizzled receptors and inhibits canonical WNT signaling	not specified	37 patients with locally recurrent or metastatic BC	Phase Ib, completed
NCT02254005	Single Dose Escalation Study of Bivatuzumab Mertansine in Female Patients With CD44v6 Positive Metastatic Breast Cancer	Bivatuzumab is an mAb anti-CD44v6	not specified	24 patients with CD44v6-positive metastatic BC	Phase I, completed
NCT02915445	EpCAM CAR-T for Treatment of Nasopharyngeal Carcinoma and Breast Cancer	EpCAM CAR-T are T cells with the chimeric antigen receptor (CAR-T) that recognize EpCAM	Persistence of EpCAM-positive circulating tumor cells	30 patients with nasopharyngeal carcinoma or BC	Phase I, unknown
NCT02776917	Study of Cirmtuzumab and Paclitaxel for Metastatic or Locally Advanced, Unresectable Breast Cancer	Cirmtuzumab is a monoclonal antibody that binds to receptor-tyrosine-kinase like orphan receptor 1 (ROR1)	ALDH1 and CD133 expression were evaluated in primary tumor samples before and after treatment	22 patients with metastatic, or locally advanced, unresectable BC	Phase I, active
NCT00645333	Phase I/II Study of MK-0752 Followed by Docetaxel in Advanced or Metastatic Breast Cancer	MK-0752 is a Gamma-secretase inhibitors	In tumor biopsies, the markers CD44, CD24, and ALDH1 were evaluated	30 patients with advanced BC	Phase I/II, completed
NCT01149356	RO4929097 And Exemestane in Treating Pre- and Postmenopausal Patients With Advanced or Metastatic Breast Cancer	RO4929097 is a gamma-secretase inhibitors. Exemestane is an aromatase inhibitor	not specified	15 patients with estrogen receptor-positive metastatic BC	Phase I, completed
NCT01876251	A Study Evaluating The PF-03084014 In Combination With Docetaxel In Patients With Advanced Breast Cance	PF-03084014 is a selective gamma-secretase inhibitor	not specified	30 advanced BC	Phase I, completed
NCT02299635	A Study Evaluating PF-03084014 In Patients With Advanced Breast Cancer With Or Without Notch Alterations	PF-03084014 is a selective gamma-secretase inhibitor	Alterations in Genes, Proteins, and RNAs Relevant to the Notch Signaling Pathway	19 advanced triple receptor-negative BC	Phase II, completed
NCT01351103	A Study of LGK974 in Patients With Malignancies Dependent on Wnt Ligands	LGK974 is an inhibitor of Wnt signaling and PDR001 (anti-PD-1)	not specified	185 patients with solid malignancies (TNBC)	Phase I, Recruiting
NCT01861054	Pilot Study to Evaluate Safety & Biological Effects of Orally Administered Reparixin in Early Breast Cancer	Reparixin is inhibitor of the CXCR1 receptor	In tissue samples, the activity of ALDH1 and markers such as CD44, CD24, Snail, Twist, and Notch were measured	20 patients with early BC	Phase II, completed
NCT02370238	A Double-blind Study of Paclitaxel in Combination With Reparixin or Placebo for Metastatic Triple-Negative Breast Cancer (FRIDA)	Reparixin is inhibitor of the CXCR1 receptor	In Metastatic tissue samples, the expression CD24, CD44, and ALDH1 was evaluated	194 patients with metastatic TNBC	Phase II, completed
NCT01118975	Vorinostat and Lapatinib in Advanced Solid Tumors and Advanced Breast Cancer to Evaluate Response and Biomarkers	Vorinostat is a histone deacetylase inhibitor. Lapatinib is a dual EGFR and HER2 inhibitor	Biomarkers of EMT and BCSCs	12 patients HER2-positive metastatic BC	Phase II, completed
NCT01868503	Lapatinib Ditosylate and Radiation Therapy in Treating Patients With Locally Advanced or Locally Recurrent Breast Cancer	Lapatinib Ditosylate and Radiation Therapy	BCSC ratio and gene expression	Seven advanced or recurrent in BC	Phase II, completed
NCT00524303	Lapatinib +/− Trastuzumab In Addition To Standard Neoadjuvant Breast Cancer Therapy	Lapatinib in combination with a standard neoadjuvant chemotherapy including 5FU, Epirubicin, Cyclophosphamide and Paclitaxel	not specified	100 patients with invasive BC overexpressing HER2	Phase II, completed
NCT02876302	Study Of Ruxolitinib (INCB018424) With Preoperative Chemotherapy For Triple-Negative Inflammatory Breast Cancer	Ruxolitinib blocks the IL6/JAK/Stat pathway	Distribution of the CD44^+^/CD24^−^ stem cell population and the inhibition of JAK and pStat3 expression	23	Phase II, Active
NCT00949013	S9313B Study of Tumor Tissue Samples from Women With Early-Stage Breast Cancer Enrolled on Clinical Trial SWOG-9313	Doxorubicin and cyclophosphamide	In tissue samples, ALDH1 was measured	1600 patients early-stage BC	Completed

mAb: monoclonal antibody; BC: breast cancer; BCSC: breast cancer stem cell; EMT: epithelial–mesenchymal transition; ALDH: aldehyde dehydrogenase; CXCR: CXC chemokine receptor; HER2: human epidermal growth factor receptor 2; VEGF: vascular endothelial growth factor receptor; EGFR: human epidermal growth factor receptor.

## Data Availability

No new data were created or analyzed in this study. Data sharing is not applicable to this article.

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
