# Peer review of "Preclinical and Clinical Trials of New Treatment Strategies Targeting Cancer Stem Cells in Subtypes of Breast Cancer"

_cells, 2023, doi:10.3390/cells12050720_

Round 1
Reviewer 1 Report
The manuscript entitled "Preclinical and clinical trials of new treatment strategies targeting cancer stem cells in subtypes of breast cancer " it is interesting, but the English style must be considered, which must be corrected. Some specific suggestions also concern the part of the abstract in which the authors say :"...of small cells known as breast cancer stem cells". In this case the authors speaking also of other types of cancer must limit themselves to mentioning the cancers stem cells. Even in the abstract, English needs to be revised. At the conceptual level in the lines 165-175 several existing methodologies to isolate/characterize the CSCs are missing; I strongly suggest to read and cite the following article DOI: 10.3390/molecules26092615 and also deepen with further ones. The way in which the sections discussing the various compounds that target CSCs are presented is not right, so please reformat them and organize them into numbered subsections.
Reviewer 2 Report
In the Manuscript “ Preclinical and clinical trials of new treatment strategies targeting cancer stem cells in subtypes of breast cancer” by Landeros et al., the authors summarized important and clinically relevant findings related to breast cancer stem-like cells (BCSC), roles and characterization of their surface biomarkers and active signaling pathways associated with the acquisition of the stemness characteristics. Moreover, the authors also covered preclinical and clinical studies that focus on targeting BCSCs. This is an important summary of the literature since targeting BCSCs is a promising therapeutic strategy against breast cancer. Although the Manuscript is mostly well written and well referenced, it needs to be modified in order to be acceptable for publication.
Specific comments:
1. The authors use term “cancer stem cells” throughout the manuscript. Is “cancer stem-like cells” more appropriate?
2. The Manuscript is missing CD90 as a marker of breast cancer stem-like cells.
3. The authors describe EpCAM as BCSC marker (line 275), but state that EMT (and decrease in EpCAM) results in acquisition of CSC phenotype (line 133-135). That is contradictory and should be addressed.
4. The authors describe Cd44+/Cd24- cells as BCSCs (line 142), however infiltrated immune cells can also be Cd44+. Adding Cd45- in the description (Cd44+/Cd24-/Cd45-) might be beneficial.
5. Lines 149-157: The authors describe interactions between CSCs and tumor microenvironment as unidirectional (microenvironment can shape CSCs). It is well documented that CSCs can shape microenvironment to support both primary tumor and metastatic growth.
6. Line 174: Xenograft is not the only in vivo tumorigenesis assay. The orthotopic model should be included.
7. Line 215: Does use of HA-coated nanoparticle carriers of chemotherapeutic drugs have side effects? Does it induce apoptosis in non-tumor CD44+ cells, for example in immune cells?
8. Line 263: The authors often use an expression “a group of researches” instead of citing the names when referencing the work of others. That should be corrected.
9. Line 295: HER2 is a membrane protein, but it is not BCSC marker. That should be clarified.
10. Line 543: The title of the section is “Immunotherapy targeting BCSCs”. However, immune therapy is defined as the treatment that activates or suppresses immune system to fight the disease. All drugs listed in this section are monoclonal antibody based targeted therapies, such as Bevacizumab, a monoclonal antibody targeting the vascular endothelial growth factor (VEGF) receptor, or Trastuzumab (HER2-monoclonal antibody), or Adecatumumab (EpCAM antibody). If use of immunotherapy term is necessary in the subtitle, it should be used as “passive immunotherapy”.
Round 2
Reviewer 1 Report
the authors have correctly followed the comments of the reviewers and corrected what was requested